# Hyperthermia and Serotonin: The Quest for a “Better Cyproheptadine”

**DOI:** 10.3390/ijms23063365

**Published:** 2022-03-20

**Authors:** Georg A. Petroianu

**Affiliations:** College of Medicine & Health Sciences, Khalifa University, Abu Dhabi 127788, United Arab Emirates; georg.petroianu@ku.ac.ae

**Keywords:** serotonin syndrome, malignant neuroleptic syndrome, 5-HT receptors, dopamine, cyproheptadine

## Abstract

Fine temperature control is essential in homeothermic animals. Both hyper- and hypothermia can have deleterious effects. Multiple, efficient and partly redundant mechanisms of adjusting the body temperature to the value set by the internal thermostat exist. The neural circuitry of temperature control and the neurotransmitters involved are reviewed. The GABAergic inhibitory output from the brain thermostat in the preoptic area POA to subaltern neural circuitry of temperature control (Nucleus Raphe Dorsalis and Nucleus Raphe Pallidus) is a function of the balance between the (opposite) effects mediated by the transient receptor potential receptor TRPM2 and EP3 prostaglandin receptors. Activation of TRPM2-expressing neurons in POA favors hypothermia, while inhibition has the opposite effect. Conversely, EP3 receptors induce elevation in body temperature. Activation of EP3-expressing neurons in POA results in hyperthermia, while inhibition has the opposite effect. Agonists at TRPM2 and/or antagonists at EP3 could be beneficial in hyperthermia control. Activity of the neural circuitry of temperature control is modulated by a variety of 5-HT receptors. Based on the theoretical model presented the “ideal” antidote against serotonin syndrome hyperthermia appears to be an antagonist at the 5-HT receptor subtypes 2, 4 and 6 and an agonist at the receptor subtypes 1, 3 and 7. Very broadly speaking, such a profile translates in a sympatholytic effect. While a compound with such an ideal profile is presently not available, better matches than the conventional antidote cyproheptadine (used off-label in severe serotonin syndrome cases) appear to be possible and need to be identified.

## 1. Introduction

Fine temperature control is essential in homeothermic animals. Both hyper- and hypothermia can have deleterious effects. Multiple, efficient and partly redundant mechanisms of adjusting the body temperature to the value set by the internal thermostat exist. Hyperthermia can be mitigated or avoided by:decreasing thermogenesis ↓ (reducing metabolic rate and brown adipose tissue (BAT) catabolism). BAT is richly innervated by sympathetic efferent fibers and β3 receptor activation induces BAT thermogenesis [1];diverting heat to the periphery (cholinergic vasodilation);increasing cooling (sweat production by the sudoriparous cholinergic glands and evaporation);decreasing motor activity (↑ parasympathetic drive).

While certainly an oversimplification, these effects can be viewed as a consequence of a decrease in sympathetic output to the periphery with a concomitant increase of parasympathetic (vagal) influence. The opposite holds for hypothermia avoidance, where an increase in sympathetic output is beneficial.

The preoptic area (POA), located in the anterior hypothalamus, is generally viewed as the brain thermostat. Input to the POA comes both from somatosensory thermo(re)ceptors in the periphery (body temperature) and thermo(re)ceptors in the POA itself (core temperature) [2]. Output from the POA is mainly–but not only–inhibitory GABA*ergic* directed towards the Sympatho-Motor Command System located in the Nucleus Raphe Dorsalis (NRD).

POA thermoceptors are transient receptor potential (TRP) (mostly) cell membrane channels; when activated, TRP channels open, allowing cation influx and activation (depolarization) of the cell. The dominant TRP channel expressed in the POA is member two of the M-(melastatin) eight-member subfamily [3,4]. The M from melastatin is related to the fact that the first member of this receptor subfamily (TRPM1) was isolated from melanoma tissue [3,5,6]. The numbering of the members corresponds to the order of their discovery. A must read review of the TRMP2 topic was recently published by Ali et al. [7].

Preoptic area (POA): The output from the POA is the result of the balance between TRPM2 and EP3 prostaglandin receptors mediated effects.

TRPM2 (activation) protects against body temperature elevation by mediating body cooling. Activation of TRPM2-expressing neurons in POA results in hypothermia, while inhibition has the opposite effect [8,9].

Conversely, EP3 receptors induce elevation in body temperature. Activation of EP3-expressing neurons in POA results in hyperthermia, while inhibition has the opposite effect (Figure 1).

Must read reviews on the topic, such as the ones authored by Trevor and Barnes and by Voronova, exist [10,11].

## 2. TRPM2-Activators

ADP-ribose is the intracellular endogenous agonist at TRPM2; it is generated as a defense response to i.a. oxidative stress. Reversible ribosylation of the TRPM2 protein opens calcium channels and induces a cation current [12,13,14]. Kheradpezhouh et al., 2014 presented data suggesting that exposure of hepatocytes to acetaminophen (N-acetyl-p-aminophenol; paracetamol), results in activation of TRPM2 channels and induces a cation current similar to that activated by oxidative stress (hydrogen peroxide; H_2_O_2_) or the intracellular application of ADP-ribose [15].

The therapeutic plasma concentration for acetaminophen is between 5 and 20 µg/mL (33 to 132 µm) [16]. The brain: plasma ratio is ≈1:5 in humans [17].

The antipyretic efficacy of acetaminophen (paracetamol, a compound lacking significant cyclooxygenase inhibitory ability), is likely to be associated with activation of TRPM2 channels in POA. At the same time TRPM2 channels mediate liver injury induced by acetaminophen toxicity.

## 3. EP3-Activators

PGE2 (dinoprostone) is the final mediator that triggers fever by acting directly on EP3 prostaglandin receptors (EP3) expressed on POA neurons [18]. PGE2 has extreme high affinity (dissociation constant K_d_ ≈ 0.3 nm) for EP3. EP3 is an inhibitory GPCR prostanoid receptor (Gi, inhibiting adenyl cyclase). EP3-deficient mice as well as mice selectively deleted of EP3 expression in the brain’s preoptic nucleus fail to develop fever in response to endotoxin (i.e., bacteria-derived lipopolysaccharide) or the host-derived regulator of body temperature, IL-1β (endogenous pyrogen). These findings indicate that activation of the EP3 receptor suppresses the GABA*ergic* inhibitory tone (inhibition of inhibition) that the preoptic hypothalamus has on thermogenic effector cells.

## 4. Exogenous TRPM2-Inhibitors

There are several known TRPM2 inhibitors, including amino-ethoxy-diphenyl borate (2-APB), amino acids (anthranilic acid), azole antifungals (clotrimazol), fenamate NSAIDs, and phenothiazine antipsychotics (chlorpromazine). However, this appears to be clinically of little or no relevance as the concentrations required to achieve even a partial block are beyond the respective safety margins. The lack of specificity is a further major limitation [19,20,21,22,23].

Toda et al., 2019 identified duloxetine (a dual inhibitor of serotonin and norepinephrine reuptake with comparable single digit nanomolar affinities for both 5-HT and NE transporters), as an inhibitor of oxidative stress-induced TRPM2 activation (open-channel blocking mechanism) [24]. Duloxetine inhibits TRPM2 independently from its ability to inhibit serotonin and norepinephrine reuptake. The therapeutic serum concentration of duloxetine ranges between 30 and 120 ng/mL (90–360 nm) [25]. The brain–blood ratio for duloxetine ranges from ≈5 to 22 [26]. Theoretically, duloxetine should favor hyperthermia development more than other reuptake inhibitors; clinically, however, this does not seem to be the case.

Conceptually peripherally acting TRPM2 inhibitors could be organo-protective (hepatotoxicity) [7]. A clear clinical benefit from centrally acting TRPM2 inhibitors has not yet been identified; such compounds favor an increase in temperature and could potentially be useful in hypothermia.

Output from the preoptic area (POA): The preoptic area of the anterior hypothalamus sends inhibitory efferent output to a number of partner and subaltern sites co-responsible for controlling body temperature;Dorsal Hypothalamic Area (DHA): Glutamatergic neurons located in the dorsal hypothalamus are under tonic inhibition from POA. When activated (disinhibited) these neurons cause an increase in body temperature via cascade activation of subaltern downstream structures (raphe pallidus). In contrast, dopaminergic input (via D2 receptors) from the dorsal hypothalamus to raphe pallidus reduces thermogenesis [27];Nucleus Raphe Pallidus (NRP): The term *raphe* refers to a ridge that separates two symmetrical parts of the body, and was used in the naming of the raphe nuclei because this collection of nuclei are clustered around the midline of the brainstem. They are considered part of the reticular formation. The raphe nuclei are the primary location in the brain for serotonin production, and the serotonin synthesized here is distributed throughout the entire central nervous system. While the NRP receives input from the dorsal hypothalamic area, the main input is from the Nucleus Raphe Dorsalis (NRD);Nucleus Raphe Dorsalis (NRD) represents the largest population of serotoninergic neurons in the brain. Functionally NRD is viewed as the main Sympatho-Motor Command System. Serotonergic neurons from the sympatho-motor command system [28] project to the nucleus raphe pallidus (NRP) and induce sympathetic activation [29,30,31,32,33] (Figure 2).

## 5. Neurotransmitters

Systemic administration of dopamine (DA) receptor agonists leads to falls in the body temperature [27]. Since dopamine itself has a very limited ability to cross the blood–brain barrier this is likely a peripheral effect, mediated by vasodilation. Centrally, dopamine and dopamine receptor agonists also have a protective effect against temperature increase as evidenced by development of hyperthermia associated with dopaminergic blockade or abrupt withdrawal of dopaminergic agonists. This is consistent with the model proposed here, where the nucleus raphe pallidus’s activation by serotonin is opposed by dopaminergic inhibition. Such an explanation was put forward already in 1989 by Kato and Yamawaki who stated that, “hyperthermia in neuroleptic malignant syndrome is due to the dominant effect of serotonin in the thermoregulatory center either by blocking the dopamine receptor or by enhancing the serotonin secretion.” [34]. It is accepted that increasing serotonin neurotransmitter concentration in the brain favors the development of hyperthermia (Serotonin Syndrome; SS) while dopamine depletion or antagonism has a similar effect, known as Neuroleptic Malignant Syndrome (NMS).

The activity of the brain neuro-circuitry is modulated by a large variety of serotoninergic auto- and heteroreceptors that need to be considered when deciding therapeutic approaches. Clinical treatment of SS, in addition to withdrawal of the offending agent and supportive care, involves the possible administration (off-label) of cyproheptadine, a first-generation promiscuous tricyclic compound with antihistamine, anticholinergic, antiserotonergic, and local anesthetic (sodium channel blocking) properties. Its use as an antidote in severe cases of SS is complicated by the lack of availability of a parenteral formulation and its lack of selectivity among serotonin receptors.

The lack of a parenteral formulation is relevant considering that many patients will have received activated charcoal, thus making the uptake of either orally or via NG tube administered drugs, if not difficult or impossible, certainly unreliable. Considering the variety of serotonin receptors and the spectrum of effects mediated, a more selective instrument could be advantageous. For comparison purposes, data for aripiprazole, an atypical antipsychotic, are presented.

### 5.1. 5-HT1A Receptors

Transgenic mice overexpressing 5-HT1A receptors show prolonged episodes of bradycardia, and 5-HT1A agonists induce bradycardia [35,36,37,38,39]. The 5-HT1A receptor agonists produce miosis in humans [40]. Measurement of pupil size seems to provide a valuable and sensitive index of 5-HT1A receptor function [41]. The 5-HT1A gene knockout animals showed increased fear and sympatho-activation under experimental conditions [42]. In conclusion, stimulation of 5-HT1A receptors causes central sympatho-inhibition and an increase in cardiac vagal drive [35]. In line with these findings, pharmacological and receptor knockout approaches support a role for 5-HT1A receptors in defense against hyperthermia [43,44,45,46]. Microinjection of a selective 5-HT1A receptor antagonist into the raphe pallidus attenuates 8-OH-DPAT-induced hypothermia [31], while a selective 5-HT1A receptor agonist induced hypothermia. Similarly, activation of 5-HT1A receptors in raphe pallidus inhibits leptin-evoked increases in brown adipose tissue thermogenesis [47].

Voronova, 2021 states that 5-HT1A activation, as a rule, leads to the development of hypothermia in animals at warm ambient temperature [10]. Antagonism at these receptors by cyproheptadine (Ki ≈ 60 nm) in the context of SS does not appear to be beneficial or desirable (Figure 3).

### 5.2. 5-HT2 Receptors

The group of 5-HT2 receptors are the most sensitive ones to the inhibitory effects of cyproheptadine with Ki values in the single digit nanomolar range. The dose of cyproheptadine recommended to ensure blockade of the 5-HT2 receptors for serotonin syndrome is 20 to 30 mg [48].

Activation of 5-HT2A and 5-HT2B receptors has predominantly excitatory effects. Some authors understand SS as a consequence of excessive activation of 5-HT2A receptors [49]. The available evidence supports the view that activation of these receptors is associated with hyperthermia, while inhibition of the same favors hypothermia [10,50,51].

The 5-HT2C receptors (Gq) are structurally similar to 5-HT2A receptors, and the two coexist in many brain regions and on the same neurons. Functionally, 5-HT2A and 5-HT2C are mostly (but not always) antagonists. While they generally play opposing facilitative and inhibitory roles, this does not seem to be the case in thermoregulation [52].

The activation of 5-HT2C inhibits neurotransmitter (dopamine) release, thus reducing dopaminergic inhibition of the NRP and increasing activation of sympathetic structures. The activation of 5-HT2C thus inhibits vagal activity and favors hyperthermia [36,53]. Antagonism (5-HT2C inhibition) is, however, apparently not so sufficient as to induce hypothermia (Figure 4).

Antagonism at these receptors by cyproheptadine appears to be beneficial.

### 5.3. 5-HT_3_ Receptors

The 5-HT3 receptors are the only ionotropic serotonin receptors. Activation of central 5-HT3 receptors are effective in hypothermia induction due to marked decrease in thermogenesis and increase in heat loss. The implication of central 5-HT3 receptors in thermoregulation and the interaction with 5-HT1A receptors was reported [10,54]; a centrally administered 5-HT3 agonist dose-dependently reduced temperature [55]. Voronova, 2021 concludes that, “all available data indicate that central 5-HT3 receptors activation leads to a decrease in body temperature of a warm-blooded organism, and this occurs due to a decrease in heat production and an increase in heat loss.” [10] (Figure 5).

Antagonism at these receptors by cyproheptadine (Ki ≈ 230 nM) does not appear to be beneficial.

### 5.4. 5-HT4 Receptor

Stimulation of 5-HT4 receptors facilitates cholinergic neurotransmission at many sites, including in the brain. The activation of 5-HT4 receptor has an excitatory effect; these receptors exert both a tonic and a phasic, positive, frequency-related control on NRD 5-HT firing activity [56,57]. Inhibition of these receptors reduces the NRD5-HT firing activity. The 5-HT4 receptor agonists and antagonists increase and decrease 5-HT cell firing, respectively [58,59].

Antagonism at these receptors by cyproheptadine–if exerted–could be beneficial. No information concerning the cyproheptadine effect at 5-HT4 receptors is available to us.

### 5.5. 5-HT5 Receptors

The 5-HT5 receptor occurs in brain areas that are implicated in learning and memory [60]. These Gi protein coupled receptors are poorly explored due to lack of selective ligands [61].

No information concerning the cyproheptadine effect at 5HT5 is available to us.

### 5.6. 5-HT6 Receptors

The 5-HT6 receptors, expressed almost exclusively in the brain, are Gs protein coupled and mediate excitatory neurotransmission. It was recognized that 5-HT6 receptors modulate primarily GABA and glutamate levels, influencing the secondary release of other neurotransmitters [62].

The Trevor Sharp and his group at Oxford found pharmacological evidence for 5-HT6 receptor modulation of serotoninergic neuron firing in vivo. The group investigated the effect of intravenous administration of high affinity and selectivity 5-HT6 agonists and antagonists, on the firing of 5-HT neurons in the NRD in vivo. The Wyeth–Ayerst 5-HT6 receptor agonist WAY-181187 caused a dose-dependent increase in the 5-HT neuron firing rate. In contrast, the Glaxo–Smith–Kline antagonist SB-399885 caused a dose-dependent decrease in the 5-HT neuron firing rate, an effect reversed by WAY-181187 [59]. They conclude that, “5-HT4 and 5-HT6 receptors might act in concert to provide a homeostatic positive feedback control of 5-HT neurons, whereas other 5-HT receptor subtypes, including 5-HT1A and 5-HT1B receptors, provide a balancing negative feedback control.”

Taken together, these findings seem to indicate that antagonism at this receptor by cyproheptadine might be desirable (Figure 6).

### 5.7. 5-HT7 Receptors

Pharmacological and receptor knockout approaches support a role for 5-HT7 receptors in fine-tuning of homeostatic regulation of body temperature [43,44,45,46,63]. Pharmacological evidence supports a role for serotonergic systems, acting via 5-HT7 receptors, in thermoregulatory cooling [45,63,64].

Landry et al. (2006) point out that stimulation of 5-HT7 receptors primarily results in hypothermia, a view confirmed by Naumenko’s group [65,66]. The effect is similar to that achieved by stimulation of 5-HT1A receptors (Figure 7).

## 6. Cyproheptadine

Based on the aforementioned, the ideal antidote for SS appears to be an antagonist at the even receptor subtypes and an antagonist at the uneven ones (with the exception of 5-HT5, for which no data are presently available). A compound with such a profile does not exist, or at least is unknown to us. Any drug used as an antidote will need to be a compromise solution. The relative importance of the various receptor subtypes in temperature control is difficult to estimate, but probably an atypical antipsychotic profile (aripiprazole-like) might be a reasonable approximation of desirable profile. Cyproheptadine is an antagonist at most (probably all) serotonin receptor subtypes; while antagonism at *even* receptor subtypes is desirable, such an effect at uneven receptors is probably less so. Aripiprazole is an antagonist at serotonin receptor subtypes 2A and 6 and a partial agonist at 1A and weak agonist at 2C and 7 (Figure 8 and Table 1).

## 7. Conclusions

The GABAergic output from the brain thermostat in the preoptic area (POA) to subaltern neural circuitry of temperature control (Nucleus Raphe Dorsalis and Nucleus Raphe Pallidus) is inhibitory. Its magnitude is a function of the balance between the (opposite) effects mediated by the transient receptor potential receptor TRPM2 and EP3 prostaglandin receptors. Activation of TRPM2-expressing neurons in POA favor hypothermia, while inhibition has the opposite effect. Conversely, EP3 receptors induce elevation in body temperature. Activation of EP3-expressing neurons in POA results in hyperthermia, while inhibition has the opposite effect. Agonists at TRPM2 and/or antagonists at EP3 could be beneficial in hyperthermia control. Activity of the neural circuitry of temperature control is modulated by a variety of 5-HT receptors. Based on the theoretical model presented the “ideal” antidote against serotonin syndrome hyperthermia appears to be an antagonist at the 5-HT receptor subtypes 2, 4 and 6 and an agonist at the receptor subtypes 1, 3 and 7. Very broadly speaking such a profile translates into a sympatholytic effect. While a compound with such an ideal profile is presently not available, better matches than the conventional antidote cyproheptadine appear to be possible and need to be identified.

## Figures and Tables

**Figure 1 ijms-23-03365-f001:**
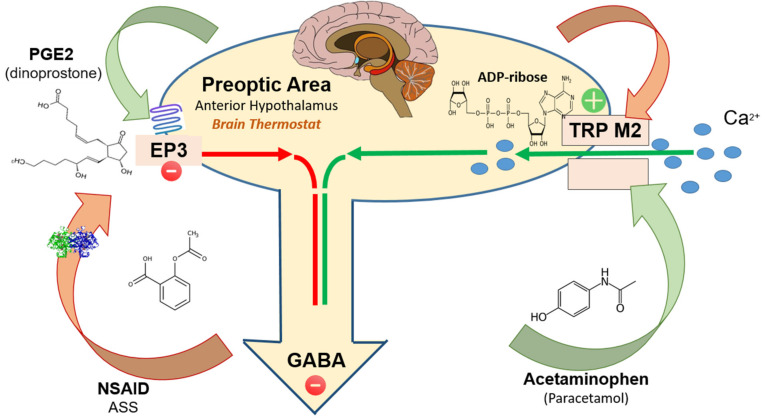
The output from the POA is the result of the balance between TRPM2 and EP3 prostaglandin receptor mediated effects. TRMP2 activation by the endogenous intracellular agonist ADP-ribose increases the POA neuronal activity, i.e., GABAergic inhibition of downstream subaltern structures. Exogenous TRP M2 agonists (acetaminophen and possibly metamizole) have a similar effect. EP3 activation by the endogenous agonist PGE2 (dinoprostone) reduces POA neuronal activity i.e., reduces GABAergic inhibition (disinhibition) of downstream subaltern structures. NSAIDs by inhibiting cyclo-oxygenases reduce the availability of PGE2 and thus the activation of EP3. TRMP2 antagonists are not (yet) in clinical use.

**Figure 2 ijms-23-03365-f002:**
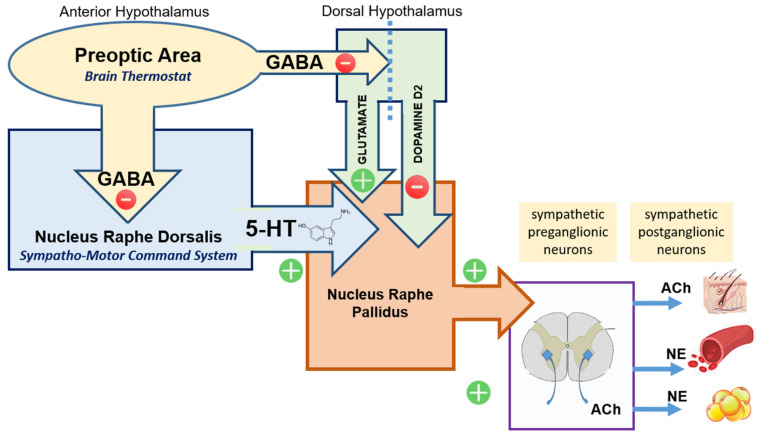
The preoptic area (POA) of the anterior hypothalamus (area) sends GABAergic inhibitory efferent output to a number of partner and subaltern sites co-responsible for controlling body temperature. Glutamatergic neurons in the dorsal hypothalamic area (DHA) are under such inhibitory control. When disinhibited they activate the nucleus raphe pallidus (NRP) that subsequently activates sympathetic neurons. NRP is also under inhibitory dopaminergic control originating from the DHA. The nucleus raphe dorsalis (NRD; Sympatho-Motor Command System) is also tonically inhibited by POA. When disinhibited it allows serotoninergic activation of the nucleus raphe pallidus (NRP) with subsequent sympathetic activation. Direct activation of sympathetic neurons (bypassing the NRP) is also possible. An increase in sympathetic output favors heat generation (BAT catabolism) and reduces heat loss (peripheral vasoconstriction).

**Figure 3 ijms-23-03365-f003:**
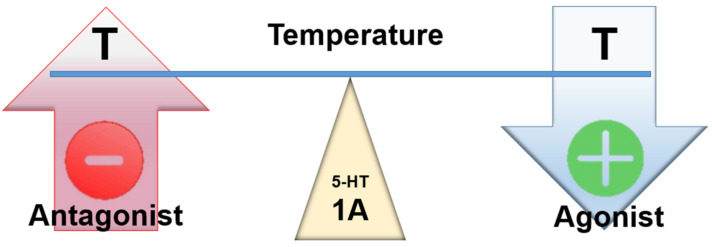
Assumed effect of centrally acting 5-HT1A agonists and antagonists on temperature.

**Figure 4 ijms-23-03365-f004:**
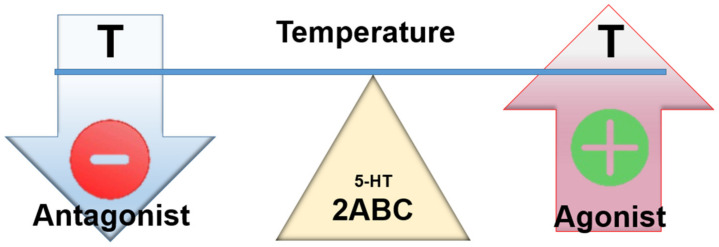
Assumed effect of centrally acting 5-HT2 agonists and antagonists on temperature.

**Figure 5 ijms-23-03365-f005:**
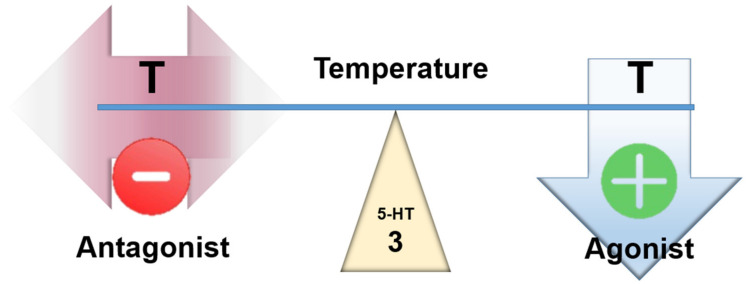
Assumed effect of centrally acting 5-HT3 agonists and antagonists on temperature.

**Figure 6 ijms-23-03365-f006:**
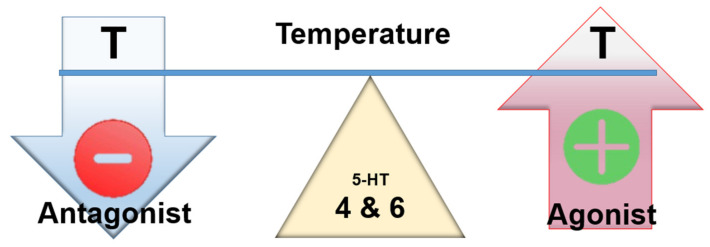
Assumed effect of centrally acting 5-HT4 and 6 agonists and antagonists on temperature.

**Figure 7 ijms-23-03365-f007:**
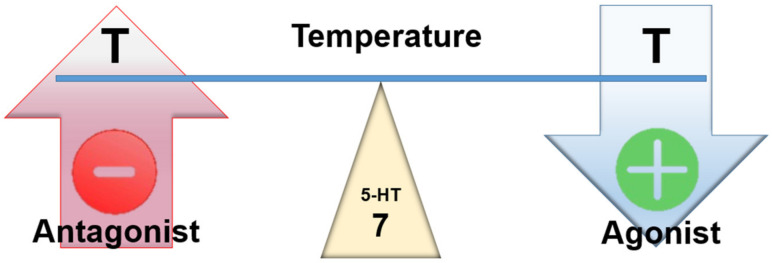
Assumed effect of centrally acting 5-HT7 agonists and antagonists on temperature.

**Figure 8 ijms-23-03365-f008:**
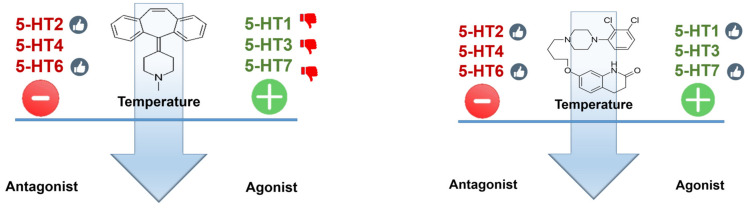
Cyproheptadine (**left**
**panel**) is an antagonist at most (probably all) serotonin receptor subtypes; while antagonism at even receptor subtypes is desirable (thumbs-up), such an effect at *uneven* receptors is probably less so (thumbs-down). Aripiprazole (**right panel**) is an antagonist at serotonin receptor subtypes 2A and 6 and a partial agonist at 1A (Emax ≈ 70% of 5-HT effect) and very weak agonist at 2C and 7 (Emax ≈ 10%).

**Table 1 ijms-23-03365-t001:** Cyproheptadine (antagonist), aripiprazole (mixed effects) and serotonin (endogenous agonist) approximate affinities (Ki nM) for serotonin receptor subtypes and the serotonin transporter (SERT). The interpretation of such data, especially concerning partial or weak agonism is difficult, as the effect (Emax as percentage of serotonin effect) of the exogenous ligand depends on the site concentration of the endogenous ligand. The origin of affinity values is the PDSP Ki database.

	Cyproheptadine 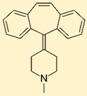	Aripiprazole 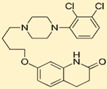	Serotonin 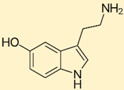
5-HT1A	60	1–10partial agonist	4
5-HT2A	2	4–60antagonist	12
5-HT2B	2	1antagonist	10
5-HT2C	2	15weak agonistEmax ≈ 10%	5
5-HT3	230	500–700	200–600
5-HT4	Not available	No effect	120
5-HT5	Not available	No effect	200–300
5-HT6	140	140–700weak antagonist	60–120
5-HT7	120	40weak agonist	2–10
SERT	4000	100 (IC_50_)	500

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
