# Peer review of "Hyperthermia and Serotonin: The Quest for a “Better Cyproheptadine”"

_ijms, 2022, doi:10.3390/ijms23063365_

Round 1
Reviewer 1 Report
IJMS Review Manuscript Hyperthermia and Serotonin: The Quest for a “Better Cyproheptadine” by Georg A Petroianu
In this review, the author is interested in the mechanisms of regulation of the body temperature focusing on serotonin and serotonin syndrome.
The neural circuitry of temperature control is first reviewed with the GABAergic inhibitory output from the preoptic area, the brain thermostat location targeting the secondary neural circuitry of temperature control in the raphe (Nucleus Raphe Dorsalis and Nucleus Raphe Pallidus). The author then summarizes the need for a balance between the effects mediated by the transient receptor potential receptor TRPM2 and the prostaglandin receptors EP3. This control implicates the activation of TRPM2-expressing neurons to induce hypothermia. Conversely, activation of the EP3 receptors increases body temperature. The author concludes that agonists at TRPM2 and/or antagonists at EP3 could be beneficial in hyperthermia control.
Then, the neurotransmitter serotonin involved in raphe circuits are reviewed. The author describes how activity of the neural circuitry of temperature control is modulated by a variety of 5-HT receptors. The author provides a theoretical model concluding that molecules with antagonist properties at the 5-HT2A-C, 4 and 6 and agonist properties at the 5-HT1A, 3 and 7 receptors could be beneficial to prevent hyperthermia.
This rather focused review could be of interest for the reader. There are however some issues that should first be clarified.
Although cyproheptadine is described as lacking selectivity among serotonin receptors, it used by the author as a reference compound to conteract the serotonin syndrome. By contrast the reason why aripiprazole is compared in Table 1 is not mentioned but should be provided. In addition, the affinity values in the text and table 1 do not match, and the origin of affinity values of table 1 is not specified; it seems that the values for cyproheptadine are taken from PDSP database but values for Aripiprazole do not completely match PDSP, in particular affinity values for 5-HT2A and 5-HT2C receptors are lower, for 5-HT2B higher. It is absolutely necessary to provide the origin of these values with their correct quote.
A similar issue concerns the Ki values given in the text for 5-HT4 and 5-HT5 receptors, which are not attributed to any ligand, do not match values in table. This should be clarified, completed, and references should be given.
Minor
-The legend for Fig8 seems to be separated in two parts
-The lines 257-258 seems to be with larger fonts
- Serotonin (5-HT) is sometime written 5HT; this should be corrected.
Author Response
Manuscript ID: ijms-1648919
Dear Editor,
Dear Mrs. Wang,
Thank you for considering -contingent on revision- our above-mentioned manuscript for publication.
Please convey our thanks to the reviewers; we appreciate the effort involved in reviewing and their constructive comments. The suggestions made contribute to increasing the quality of this manuscript and of future submissions.
Please find below a list of the comments made and changes requested by the reviewers followed by our answer (blue font) and/or the description of the specific action taken in order to comply (red font).
Attached is the revised manuscript with changes highlighted.
Yours sincerely
Georg Petroianu MD, PhD, FCP, FAPE
Reviewer 1
Q1. This rather focused review could be of interest for the reader.
Thank you very much indeed for the kind comment; truly appreciated.
Q2. Although cyproheptadine is described as lacking selectivity among serotonin receptors, it used by the author as a reference compound to counteract the serotonin syndrome
In clinical practice, in severe cases of Serotonin Syndrome when supportive measures are not sufficient, cyproheptadine is the most commonly used anti-serotoninergic agent. While cyproheptadine is FDA approved, its use for Serotonin Syndrome is off-label.
This aspect was emphasized.
Q3 By contrast the reason why aripiprazole is compared in Table 1 is not mentioned but should be provided.
Point well taken.
An explanation was included.
Q4 The affinity values in the text and table 1 do not match
Thank you for noticing the mistake.
The discrepancy was corrected.
Q5 The origin of affinity values of table 1 is not specified and; it seems that the values for cyproheptadine are taken from PDSP database but values for Aripiprazole do not completely match PDSP, in particular affinity values for 5-HT2A and 5-HT2C receptors are lower, for 5-HT2B higher. It is absolutely necessary to provide the origin of these values with their correct quote
Indeed the origin of affinity values is the PDSP database; the PDSP database lists all or most available studies regarding Ki values so it provides a range and not an absolute generally accepted value. As such, a complete match is not always possible.
For aripiprazole and 5-HT2A a range is now given. The PDSP Ki database is cited.
Q6 A similar issue concerns the Ki values given in the text for 5-HT4 and 5-HT5 receptors, which are not attributed to any ligand, do not match values in table. This should be clarified, completed, and references should be given.
Thank you for noticing the mistake.
The values were removed.
Reviewer 2
Q7 The GABAergic inhibitory output from the brain thermostat in the preoptic area POA to subaltern neural circuitry of temperature control (Nucleus Raphe Dorsalis and Nucleus Raphe Pallidus) is a function of the balance between the (opposite) effects mediated by the transient receptor potential receptor TRPM2 and EP3 prostaglandin receptors. This needs to be restructured in a better way.
Point well taken
The sentence was split; it now reads: The GABAergic output from the brain thermostat in the preoptic area POA to subaltern neural circuitry of temperature control (Nucleus Raphe Dorsalis and Nucleus Raphe Pallidus) is inhibitory. Its magnitude is a function of the balance between the (opposite) effects mediated by the transient receptor potential receptor TRPM2 and EP3 prostaglandin receptors.
Q8 Adding brackets for the abbreviations will make it easier for readers to follow.
Point well taken
Brackets were inserted.
.
Q9 Activity of the neural circuitry of temperature control is modulated by a variety of 5-HT receptors. First hand use of abbreviations needs to be looked at and the author is advised for a consistent usage of abbreviations throughout the manuscript.
Point well taken
The manuscript was revised for consistent usage of abbreviations
Q10 Based on the theoretical model presented the “ideal” antidote against serotonin syndrome hyperthermia appears to be an antagonist at the 5-HT receptor subtypes 2, 4 and 6 and an agonist at the receptor subtypes 1, 3 and 7. The author is unable to explain the meaning in this sentence. In general, sentence should be crisp with a clear meaning but in the present manuscript I can see long sentences that are difficult to follow and this is a serious issue that needs to be addressed.
Point well taken
The sentence was split; it now reads: Based on the theoretical model presented the “ideal” antidote against serotonin syndrome hyperthermia appears to be an antagonist at the 5-HT receptor subtypes 2, 4 and 6 and an agonist at the receptor subtypes 1, 3 and 7.
Q11 GABAergic inhibition (disinhibition). What does authors meant by disinhbition?
GABAergic neurons are inhibitory. GABA receptors on GABA neurons (auto-receptors) by inhibiting the inhibitory neuron have an overall disinhibiting effect
Q12 The manuscript covers important topics but lacks clarity in few aspects and need to improve in that context.
Thank you very much indeed for the kind comment and guidance.
We rephrased confusing sentences.

Reviewer 2 Report
The article entitled “Hyperthermia and Serotonin: The Quest for a “Better Cypro- 2 heptadine” provides important insights into various concepts but need improvement before it can be accepted for publication.
- The GABAergic inhibitory output from the brain thermostat in the preoptic area POA to subaltern neural circuitry of temperature control (Nucleus Raphe Dorsalis and Nucleus Raphe Pallidus) is a function of the balance between the (opposite) effects mediated by the transient receptor potential receptor TRPM2 and EP3 prostaglandin receptors. This needs to be restructured in a better way.
- preoptic area POA; adding brackets for the abbreviations will make it easier for readers to follow.
- Activity of the neural circuitry of temperature control is modulated by a variety of 5-HT receptors. First hand use of abbreviations needs to be looked at and the author is advised for a consistent usage of abbreviations throughout the manuscript.
- Based on the theoretical model presented the “ideal” antidote against serotonin syndrome hyperthermia appears to be an antagonist at the 5-HT receptor subtypes 2, 4 and 6 and an agonist 20 at the receptor subtypes 1, 3 and 7. The author is unable to explain the meaning in this sentence. In general, sentence should be crisp with a clear meaning but in the present manuscript I can see long sentences that are difficult to follow and this is a serious issue that needs to be addressed.
- GABAergic inhibition (disinhibition). What does authors meant by disinhbition?
- The manuscript covers important topics but lacks clarity in few aspects and need to improve in that context.

Author Response

(The authors gave the same response as above.)

Round 2
Reviewer 1 Report
IJMS Revised Review Manuscript Hyperthermia and Serotonin: The Quest for a “Better Cyproheptadine” by Georg A Petroianu
In this revised review, the author improved substantially the initial manuscript.
Reviewer 2 Report
All the comments have been well addressed and manuscript can now be accepted for publication.